# Cardio-Ankle Vascular Index in the Persons with Pre-Diabetes and Diabetes Mellitus in the Population Sample of the Russian Federation

**DOI:** 10.3390/diagnostics11030474

**Published:** 2021-03-08

**Authors:** Alexei N. Sumin, Natalia A. Bezdenezhnykh, Andrey V. Bezdenezhnykh, Galina V. Artamonova

**Affiliations:** Laboratory of Comorbidity in Cardiovascular Diseases, Federal State Budgetary Scientific Institution “Research Institute for Complex Issues of Cardiovascular Diseases”, 6 Sosnoviy Blvd, Kemerovo 650002, Russia; n_bez@mail.ru (N.A.B.); andrew22014@mail.ru (A.V.B.); artamonova@kemcardio.ru (G.V.A.)

**Keywords:** arterial stiffness, cardio-ankle vascular index, carbohydrate metabolism disorders, type 2 diabetes, prediabetes, ESSE-RF study

## Abstract

The aim of this study was to evaluate Cardio-Ankle Vascular Index (CAVI) and increased arterial stiffness predictors in patients with carbohydrate metabolism disorders (CMD) in the population sample of Russian Federation. Methods: 1617 patients (age 25–64 years) were enrolled in an observational cross-sectional study Epidemiology of Cardiovascular Diseases and Their Risk Factors in the Regions of the Russian Federation (ESSE-RF). The standard ESSE-RF protocol has been extended to measure the cardio-ankle vascular index (CAVI), a marker of arterial stiffness. Patients were divided into three groups: patients with type 2 diabetes mellitus (*n* = 272), patients with prediabetes (*n* = 44), and persons without CMD (*n* = 1301). Results: Median CAVI was higher in diabetes and prediabetes groups compared with group without CMD (*p* = 0.009 and *p* < 0.001, respectively). Elevated CAVI (≥9.0) was detected in 16.8% of diabetes patients, in 15.9% of those with prediabetes, and in 9.0% of those without CMD (*p* < 0.001). The factors affecting on CAVI did not differ in CVD groups. In logistic regression the visceral obesity, increasing systolic blood pressure (SBP) and decreasing glomerular filtration rate (GFR) were associated with a pathological CAVI in CMD patients, and age, diastolic blood pressure (DBP), and cholesterol in persons without CMD. Conclusions: the CAVI index values in the prediabetes and diabetes patients were higher than in normoglycemic persons in a population sample of the Russian Federation. Since the identified disorders of arterial stiffness in prediabetes are similar to those in diabetes, their identification is important to prevent further cardiovascular complications.

## 1. Introduction

In patients with diabetes mellitus (DM), cardiovascular diseases are the main cause of death [1,2]. Further, recent studies have shown that classic microvascular and macrovascular disorders can be detected already in pre-diabetes [3]. Prediabetes has been shown to contribute to the pathogenesis of macrovascular dysfunction, which may in part explain the increased risk of cardiovascular disease and mortality in prediabetes and type 2 diabetes [4]. Naturally, for adequate prevention of cardiovascular complications, it is necessary to identify their early precursors.

One of these markers is increased arterial stiffness [5,6]. Traditionally used to assess this parameter, the indicator carotid-femoral pulse-wave velocity has a number of limitations (lack of standardization, dependence on the operator, and on the level of blood pressure) [7]. In recent years, a new indicator has begun to be used to assess arterial stiffness the cardio-ankle vascular index (CAVI), which is devoid of the above disadvantages [8].

This index reflects the stiffness of the arterial tree from the beginning of the aorta to the ankle; the CAVI principle and the formula for its calculation were described in more detail in previous publications [9]. It is now recognized that CAVI has several unique logistic and conceptual properties: first, ease of measurement (using blood pressure cuffs placed on both arms and ankles, and a microphone on the chest, without the need for sensors on the neck or in the groin) and operator independence; second, CAVI measures the stiffness of the entire aorta (including the ascending segment), femoral, popliteal, and tibial arteries, and measures the increase in arterial stiffness from end diastole to end systole; third, CAVI is less influenced by blood pressure during measurement compared to pulse wave velocity (PWV), since CAVI is based on the stiffness parameter β, which reflects the degree of pressure-volume ratio [10]. It can be noted that the convenience of using this index in clinical practice has made it possible to expand and deepen research on this topic, which is reflected, in particular, in recently published reviews [11,12,13].

An increase in CAVI was noted in diabetic patients compared to healthy individuals. CAVI is associated with HbA1c levels [14,15]. An elevated CAVI (≥ 8.0) is independently associated with diabetes [15]. In diabetic patients, CAVI is associated with the presence of coronary artery disease [14,16] and microvascular complications (retinopathy, peripheral neuropathy, and microalbuminuria) [14,17,18]. In prospective study in patients with diabetes, the association of CAVI with the development of cardiovascular events has been shown [19].

Previous studies investigating the relationship between arterial stiffness and prediabetes were inconsistent [20,21,22,23,24]. In patients with prediabetes who were diagnosed based on impaired fasting glucose (IFG), increased pulse wave velocity was associated with IFG and HbA1c [21]. However, in another studies the detection of increased arterial stiffness in prediabetes has not been found [20,22]. Less data on CAVI values in patients with prediabetes. For example, in the *improving interMediAte RisK management* (MARK) study, the level of CAVI in the healthy and pre-diabetes groups did not differ and was increased only in the presence of diabetes [24]. On the contrary, Tsuboi et al. reported that glucose levels 1 h after a meal are associated with increased CAVI in non-diabetic subjects [23]. Therefore, the relationship between prediabetes and CAVI requires further research.

It can be concluded that there is no clear relationship between markers of dysglycemia and CAVI in subjects with normal glucose metabolism, prediabetes, and type 2 diabetes. The aim of this study was to evaluate the level of CAVI in the persons with pre-diabetes and diabetes mellitus in the population sample of the Russian Federation, as well as to investigate the factors associated with pathological CAVI in these subjects.

## 2. Subjects, Materials and Methods

The study was carried out within the framework of a multicenter observational epidemiological study ESSE-RF (Epidemiology of Cardiovascular Diseases and Their Risk Factors in the Regions of the Russian Federation) in the Kemerovo region in the period from 2012 to 2013 years [25]. The object was a random population sample of the adult male and female population of the Kemerovo region (Western Siberia, Russia) aged 25–64 years, predominantly Caucasians. The standard protocol of the ESSE-RF has been extended with additional study of cardio-ankle vascular index (CAVI). In total, 2000 people were invited to the study, and 1628 people took part in the study (the response was 81.4%). As a result, a sample of 1619 people was formed, about whom there was complete information for this study.

The study design is shown in Figure 1. From the sample of 1619 people described above, patients with carbohydrate metabolism disorders were identified: type 1 and 2 diabetes mellitus, impaired fasting glucose (IFG), impaired glucose tolerance (IGT)—a total of 318 people. Two patients with type 1 diabetes mellitus were excluded from further analysis due to the fact that patients with type 1 diabetes mellitus have obviously more pronounced changes in arterial stiffness, which are not comparable with the rest of the sample.

The remaining 1617 people were divided into three groups: group 1—patients with type 2 diabetes mellitus (*n* = 272, of which 54 patients (19.8%) with newly diagnosed type 2 diabetes mellitus); group 2—patients with prediabetes—IFG, IGT, or their combination (*n* = 44, of which 13 patients (29.5%) with newly diagnosed prediabetes); group 3—persons without any confirmed carbohydrate metabolism disorders (CMD, *n* = 1301). In these groups, clinical and anamnestic data, anthropometric characteristics, and CAVI index were analyzed depending on the presence of carbohydrate metabolism disorders. Further, to analyze vascular stiffness and related patterns, 49 patients were excluded, who had not been tested on the VaSeraVS-1000 device and had not been identified by CAVI.

In addition, 77 patients with ankle-brachial index (ABI) values less than 0.9 were also excluded from the analysis. Thus, the final sample for analyzing the patterns associated with CAVI was 1491 persons.

### 2.1. Clinical and Biochemical Data Collection

Physical examination included measurement of blood pressure, heart rate, anthropometric indicators; registration of electrocardiogram (ECG) at rest in 12 leads; and taking blood for biochemical laboratory tests.

Blood pressure was measured on the subject’s right arm with an automatic tonometer in a sitting position, after a 5-min rest. The blood pressure was measured twice with an interval of ~2 to 3 min. The analysis included the mean of two measurements. Patients were considered to have arterial hypertension if blood pressure was 140/90 mmHg and more or they were currently receiving antihypertensive treatment in accordance with the current guidelines of the European Society of Cardiology. Measurements of height and body weight were carried out using a height meter with an accuracy of 1 cm and an outdoor electronic medical scale with an accuracy of 100 g; the subject was without shoes and outerwear. Obesity was determined at a body mass index (BMI) ≥ 30 kg/m^2^, which was calculated by the formula: weight in kg/height in m^2^ (Quetelet index). Waist circumference (WC) and hip circumference (HC): the measurements were carried out in a standing position, the measuring tape was placed horizontally; for WC, the measurement point was the midpoint of the distance between the apex of the iliac crest and the lower lateral edge of the ribs. Visceral obesity was defined with a waist circumference equal to or greater than 80 cm in women and 94 cm in men. Persons who smoked 1 cigarette or more per day were considered regular smokers.

All blood samples were collected after overnight fasting of at least 12 h, and glucose, creatinine, cholesterol, triglyceride, high-density lipoprotein cholesterol (HDL-C), low-density lipoprotein cholesterol (LDL-C), and uric acid levels were determined. Laboratory methods were strictly standardized and performed on the same laboratory equipment using the same reagent kits in clinical laboratories. Glomerular filtration rate (GFR) was calculated from creatinine levels using the CKD-EPI (Chronic Kidney Disease Epidemiology Collaboration) formula.

The cardiac screening program included a survey based on a standard questionnaire consisting of 12 subsections (modules): socio-demographic data of the respondent; food habits; physical activity; smoking; alcohol consumption; health, attitude to health, and quality of life; sleep; economic conditions and work; stress; anxiety and depression; data on medical care and disability, as well as a history of diseases: angina pectoris, myocardial infarction, arterial hypertension, diabetes mellitus, and others.

### 2.2. Measurement of CAVI

Cardio-ankle vascular index was measured with the VaSeraVS-1000 device (Fukuda Denshi Company Ltd., Tokyo, Japan). This method is based on the estimate of the arterial stiffness index b, of the aorta and the iliac, femoral, and tibial arteries. CAVI was measured in study participants for the right and left lower extremities. The patient was resting in a supine position. Cuffs were placed approximately 2 cm above the intercubital fossa on the arms and 2 cm above the medial malleolus on the legs. The elbow and the heel were elevated on the special pillows to stabilize the pulse wave. The fixed cuffs did not touch the surface of the couch. ECG electrodes were placed a few centimeters above the wrist on each forearm. A small microphone was taped onto the chest. The measurements took no more than 10–15 min. The automatic calculation of this indicator is carried out on the basis of registration of plethysmograms of 4 limbs, ECG, phonocardiogram, using a special algorithm for calculations using Bramwell–Hill’s equation [10], the highest CAVI value was used for further calculations. CAVI ≥ 9.0 was defined as pathological and signified increased vascular stiffness.

### 2.3. Diagnosis of Diabetes Mellitus and Other Glycemic Disorders

Carbohydrate metabolism disorders were defined in accordance with the WHO diagnostic criteria for diabetes mellitus and other disorders of glycemia [26]. Most cases of type 2 diabetes and prediabetes were established by history and the patient’s medical records (see study design). In the case of a newly established carbohydrate metabolism disorder, the following criteria were used for diagnosis. In the absence of previously established diabetes mellitus and in the presence of borderline fasting hyperglycemia (6.1–6.9 mmol/L) or previously known prediabetes, an oral glucose tolerance test (OGTT) was performed and glycated hemoglobin was determined. Diabetes mellitus was established in the case of fasting glycemia ≥7.0 mmol/L, glycemia at 120 min of OGTT ≥11.1 mmol/L; blood glucose at random determination ≥11.1 mmol/L in the presence of typical symptoms of hyperglycemia (polydipsia, polyuria, and weakness). In the absence of symptoms of acute metabolic decompensation, the diagnosis of diabetes mellitus was established on the basis of two digits in the diabetic range, for example, double-determined blood glucose or a single determination of HbA1c + single determination of blood glucose. The level of glycated hemoglobin HbA1c ≥ 6.5% corresponded to diabetes mellitus [26].

When determining prediabetes (impaired fasting glycemia, impaired glucose tolerance) we also used the WHO 1999–2013 criteria, which are adopted in our country, and not the more stringent criteria of the American Diabetes Association. According to WHO criteria Impaired glucose tolerance (IGT) was diagnosed with fasting plasma glucose <7.0 mmol/L and 2h glucose 7.8–11.1 mmol/L. Impaired fasting glucose (IFG) has been diagnosed with fasting plasma glucose between 6.1 and 6.9 mmol / L and (if measured) plasma glucose after 2 h < 7.8. HbA1c level up to 6.0% was considered normal, HbA1c level 6.0–6.4% corresponded to prediabetes. The term prediabetes was understood as impaired fasting glucose (IFG) or impaired glucose tolerance (IGT), or a combination of both.

### 2.4. Ethics Statement

The study protocol was approved (date of approval: 08 February 2012) by the Local Ethical Committee of Research Institute for Complex Issues of Cardiovascular Diseases (Protocol No. 20120208) and was performed in accordance with the declaration of Helsinki. Patients were included in the study after they provided written informed consent.

### 2.5. Statistical Analysis

Statistical processing was using STATISTICA 8.0 and SPSS 17.0 software packages. The Shapiro–Wilk test was used to check the distribution. The data were presented as medians and quartiles (Me (LQ; UQ)), if variables were not normally distributed. The Kruskal–Wallis test was used to assess the differences between the three groups (normoglycemia, prediabetes, and diabetes). For pairwise comparison of groups, The Mann–Whitney test and χ2 (chi-square) test were used to compare two independent groups. With a small number of observations, Fisher’s exact test was used with Yates’ correction. The Bonferroni correction was used to solve the problem of multiple comparisons. Thus, taking the number of degrees of freedom into account, the critical level of significance *p* when comparing the three groups was taken equal to 0.017. Multiple linear regression analysis with stepwise selection was applied to evaluate significance correlation of CAVI with continuous parameters in groups with and without CMD. Binary logistic regression with forward selection (Likelihood Ratio) was used to assess the significant relationship between the presence of pathological CAVI and quantitative or qualitative features in groups with and without CMD. The level of critical significance (*p*) during the regression analysis was taken equal to 0.05.

## 3. Results

The main clinical and anamnestic characteristics in the groups of subjects with diabetes (*n* = 272), pre-diabetes (*n* = 44), and normoglycemia (*n* = 1301) are summarized in Table 1.

In the diabetes and pre-diabetes groups, patients were older (55.0 and 52.5 y) than in the group without CMD (46.0 y, *p* < 0.001), in the diabetes group there were fewer men (33.1%) compared to the groups with pre-diabetes and with normoglycemia (52.3% and 44.9%, *p* = 0.002). The following trend can be noted: patients with diabetes and prediabetes were comparable in most parameters, but differed from patients without CMD. Among patients of both groups with diabetes and prediabetes, there was a greater prevalence of arterial hypertension, kidney disease compared with the normoglycemia group (Table 1). Patients with diabetes were more likely to have a history of coronary heart disease and stroke in comparison with the normoglycemia group (*p* < 0.001) but were comparable to the group of prediabetes (*p* > 0.05). Patients in the diabetic and prediabetes groups had a greater smoking duration (*p* < 0.05) compared with the group without CMD, without differing from each other. At the same time, the number of smokers was smaller in both groups with CMD in comparison with the group without CMD (*p* = 0.005). The group without CMD had the highest percentage of employed (*p* = 0.001 in comparison with the group with diabetes. When analyzing anthropometric characteristics (Table 1), a similar trend was observed—patients with diabetes and prediabetes did not differ in most of the parameters, but at the same time differ from the group without CMD. The median weight and BMI in the diabetes and pre-diabetes groups was greater than in the group without BMD (*p* < 0.001 in all cases). Waist and hip circumference was significantly greater in the groups with CMD (*p* < 0.001 compared with the group without CMD). Signs of obesity (BMI ≥30 kg/m^2^) were detected in 60.3% of diabetes patients (*p* = 0.003 versus group without CMD), 50% of prediabetes patients (*p* < 0.001 versus group without CMD), and 29.1% of patients without CMD.

Visceral obesity (waist circumference > 80 cm in women and > 94 cm in men) was extremely common among patients with CMD—84.1% in diabetes group (*p* <0.001 versus group without CMD), 79.6% in prediabetes group (*p* = 0.020 versus group without CMD), and 62.2% in the group without CMD.

Analysis of volumetric sphygmography indicators supported the trend of similarity of characteristics of patients with type 2 diabetes and prediabetes with a significant difference from patients without CMD. VaSera VS-1000 examination with CAVI determination was carried out in 1586 people (Table 1). The median heart rate in the CMD groups did not differ; at the same time, it was significantly higher than in the group without CMD. Medians of systolic and diastolic blood pressure were also higher in both groups with CMD (*p* < 0.001 in both cases compared with the group without CMD). The median CAVI was higher in diabetes and pre-diabetes groups (*p* = 0.009 and *p* < 0.001, respectively, compared with group without CMD). CAVI ≥8 occurred in 40.3% of diabetic patients (*p* < 0.001 versus group without CMD), 38.6% of patients with prediabetes (*p* = 0.009 versus group without CMD) and in 23.3% of those examined without CMD. CAVI ≥9.0 was defined as pathologic. Pathologic CAVI (≥ 9.0) was detected in 16.8% of diabetes patients, in 15.9% of those with prediabetes, and in 9.0% of those without CMD (*p* < 0.001 when comparing diabetes group and group without CMD, Figure 2; Table 1).

The analysis of lipid profile indicators did not contradict the previously obtained patterns: patients with prediabetes and diabetes had higher values of total cholesterol, triglycerides, and low-density lipoprotein cholesterol (LDL-C) in comparison with individuals without CMD, while not differing among themselves (Table 1). The pre-diabetes and diabetes groups were comparable in terms of the medians of uric acid, despite the fact that patients without CMD had a significantly lower level.

The glomerular filtration rate (GFR) calculated using the CKD-EPI formula was the highest in group without CMD, compared to both groups with CMD. In the groups with diabetes and prediabetes, GFR was comparable, and significantly lower in comparison with the group without CMD (Table 1). The described trend was not observed for the creatinine level, which confirms its low sensitivity in comparison with the calculated GFR formulas in relation to the reflection of the kidneys filtration function.

According to the result of the correlation analysis conducted in the general sample, CAVI positively correlated with age, male sex, type 2 diabetes, waist circumference, visceral obesity, but not with BMI (Figure 3). CAVI also positively correlated with the level of total cholesterol, triglycerides, and LDL cholesterol. An increase in CAVI with deterioration of the filtration function of the kidneys, as evidenced by the inverse correlation with GFR CKD-EPI (Figure 3), is noteworthy as well.

A multiple regression was run to predict CAVI from age, systolic blood pressure (SBP), diastolic blood pressure (DBP), Heart Rate, WC, GFR_CKD-EPI, BMI, TG, Glucose, Cholesterol, and LDL (Table 2). In normoglycemia group age, DBP, Heart Rate, WC, and BMI statistically significantly predicted CAVI, F (5, 1162) = 82.622, *p* < 0.0001, R^2^ = 0.263. All five variables added statistically significantly to the prediction, *p* < 0.05. In CMD group only age, SBP, and BMI statistically significantly predicted CAVI, F (3, 278) = 30.90, *p* < 0.0001, R^2^ = 0.252. All three variables added statistically significantly to the prediction, *p* < 0.05.

A logistic regression was performed to ascertain the effects of variable on the likelihood that participants have pathological CAVI (≥ 9.0) in groups with CMD and with normoglycemia (Table 3). In normoglycemia subjects the logistic regression model was statistically significant, χ^2^(3) = 111.5, *p* < 0.0001. The model explained 19.9% (Nagelkerke R^2^) of the variance in pathological CAVI and correctly classified 90.9% of cases. Increasing age and DBP was associated with an increased likelihood of exhibiting pathological CAVI, but cholesterol increasing was associated with a reduction in the likelihood of exhibiting pathological CAVI in these subjects. In CMD patients the logistic regression model was statistically significant, χ^2^(4) = 35.32, *p* < 0.0005. The model explained 21.0% (Nagelkerke R^2^) of the variance in pathological CAVI and correctly classified 87.1% of cases. Patients with visceral obesity were 4.42 times more likely to exhibit pathological CAVI than patients without visceral obesity, but for general obesity, an inverse relationship was noted. Increasing SBP and decreasing GFR were associated with an increased likelihood of exhibiting pathological CAVI in CMD patients.

## 4. Discussion

In the present study, it was shown that in patients with prediabetes, CAVI values were increased to the same extent as in patients with type 2 diabetes and significantly differed from the CAVI values in individuals without CMD. The groups with diabetes and prediabetes also did not differ in factors affecting the level of arterial wall stiffness (sex, blood pressure, age, dyslipidemia, and uric acid level). In the groups with normoglycemia and CMD various associations of indicators with CAVI were noted. Pathological CAVI in normoglycemia was determined by age, blood pressure and dyslipidemia; in patients with CMD by blood pressure, visceral obesity, and decreased GFR.

Our results were somewhat unexpected and differ from those obtained earlier [24,27]. So, in the MARK study, CAVI values in patients with prediabetes were comparable to those with normoglycemia and were significantly less than in patients with diabetes [24]. Such striking differences from our results can be explained by several reasons. First, in this study, 3 criteria were used to detect prediabetes (fasting glucose, glucose tolerance test, and HbA1c level), we used only two of them (fasting glucose and impaired glucose tolerance). However, in our opinion, this could not affect the identified by us association of prediabetes and the CAVI index, since in previously published studies it was the HbA1c index that was most associated with arterial stiffness [28], including in patients with prediabetes [20,21]; therefore, its definition could only strengthen the association we identified, but not level it. Secondly, in the MARK study, a more homogeneous sample was examined persons with intermediate risk; in our sample of the epidemiological study, a wider range of subjects was represented, both with no risk and with a high level of it. This is probably why the age differences between the normoglycemic group and the CMD groups were significantly more pronounced in our study than in the MARK study. It is likely that this could have influenced the differences between the groups in these groups in the CAVI index. However, in the MARK study, as in our work, the age of patients with prediabetes and diabetes did not differ, but the CAVI values in patients with prediabetes were lower than in those with diabetes, and in ours they were comparable. Third, the influence of national and geographic characteristics cannot be ruled out. True, in this case there is no influence of racial differences, as is usually emphasized in the analysis of Asian studies on CAVI both studies were carried out in Caucasians, but the nature of nutrition and mentality of the inhabitants of Spain and southern western Siberia may well leave an imprint on the revealed patterns [29]. In any case, the revealed discrepancies in relation to prediabetes with arterial stiffness require clarification in further studies.

Other data on the association of arterial stiffness with different variants of carbohydrate metabolism disorders were revealed in the work of Namekata T. et al., [27]. The CAVI index was moderately increased in persons with prediabetes compared to those with normoglycemia and to an even greater extent in patients with diabetes. A feature of this study was the younger age of the subjects (these were employees of the company and members of their families), and most of the factors potentially affecting arterial stiffness (for example, age, systolic blood pressure, and body mass index) had similar trends they had the lowest values in patients with normoglycemia, higher in prediabetes, and most pronounced in diabetic patients. Nevertheless, it is impossible to judge the reliability of the differences in the groups with prediabetes and diabetes, since such data are not presented in the article. Based on the results of this article, it seems that the association of prediabetes with arterial stiffness is mediated by a number of concomitant factors that also affect the state of the arterial wall. However, in a study by Shen L et al. [21] after adjusting for age, sex, blood pressure, BMI, and triglycerides, brachial-ankle PWV (baPWV) was significantly higher in subjects with prediabetes (as measured by both HbA1c and IFG) compared with subjects with normoglycemia, which, apparently, indicates an independent role of the initial manifestations of CMD on arterial stiffness.

Other studies on the prediabetes association with arterial stiffness examined other indicators of the arterial wall state based on the assessment of the pulse wave velocity [22,30]. It should be noted that contradictory results were also noted in these studies. In an early study by Ohnishi H et al. [30] found significant differences between the baPWV values in the normoglycemic group and in the IFG (*p* < 0.01) and diabetic (*p* < 0.0001) groups, but there were no differences between the prediabetes and diabetes groups. The results of multiple regression analysis showed that fasting glucose was closely associated with baPWV, as well as with age and SBP. Similar results were obtained by us in the intergroup comparison; however, the differences in the regression analysis are explained by the differences in the surveyed sample (in this study a small number of patients were included, only men were examined). In the study by Chirinos JA, et al. [22], when assessing carotid-femoral PWV in groups of patients with IFG and diabetes, there was a more pronounced arterial stiffness in the IFG group compared with normoglycemia, and its even greater increase in the group with diabetes. These three groups at the same time significantly differed in age, which led, when adjusted for age and sex, to no difference between the groups with normoglycemia and IFG.

You can note, the discrepancy between the results obtained in different studies evaluating arterial stiffness in prediabetes can be due to a number of reasons. Firstly, these are the characteristics of the subjects, which were analyzed; secondly, applying various methods of measuring arterial stiffness; and, thirdly, using various adjusted variables. All of these factors make it difficult to compare results between studies. Prospective studies that adjust the underlying variables affecting arterial stiffness are needed to understand the role of different blood glucose measurements on arterial stiffness in subjects with prediabetes.

Nevertheless, the results of prospective studies can serve as indirect evidence in favor of the adverse effect of prediabetes on the state of both the arterial wall as a whole and the severity of atherosclerotic changes in it [31,32,33]. So, in a recent national cohort study in Korea, with a 12-year prospective observation, it was shown that the DM group showed the highest incidence and risk of vascular complications (cardiovascular, renal; and retinal), followed by the IFG group and the lowest risk was in group of normoglycemia [32]. It is clear that partly the intermediate risk of vascular complications in the prediabetes group could be associated with subsequent development of diabetes. However, the value of the study is different identifying patients with an increased risk of complications at the earliest possible stage, therefore identifying prediabetes makes it possible to prevent its transition to diabetes and reduce the likelihood of complications. In a meta-analysis of longitudinal follow-up studies, it was found that compared with patients with normoglycemia, coronary artery disease (CAD) patients with prediabetes on admission have a significantly higher risk of developing major adverse cardiovascular events after percutaneous coronary intervention (PCI). The potential prognostic role of prediabetes in these patients is the same regardless of study design, sample size, CAD subtype, PCI type, diabetes definition or follow-up duration, and even after adjusting for CAD severity [33].

It is clear that these studies did not assess arterial stiffness, but the study of CAVI may be useful in another aspect. Since the presence of prediabetes is an unfavorable prognostic factor, targeted measures are needed both to normalize carbohydrate metabolism and to influence other associated risk factors in the complex. In our study, it was shown that pathological CAVI in patients with CMD is associated primarily with the level of blood pressure and the presence of visceral obesity, which may be the primary targets for impact in secondary prevention programs. At the same time, the assessment of CAVI in dynamics is quite capable of being the tool that will allow to monitor the success of measures to prevent cardiovascular complications. This assumption is consistent with the data Otsuka K. et al. [34] that the incidence of cardiovascular events in CAD patients with an initially elevated CAVI after 2.9 years was significantly higher in the group without improvement in CAVI after 6 months than in the group with its improvement. These interesting results show, firstly, the possibility of improving CAVI in CAD patients and, secondly, reflect the beneficial effect of such an improvement on their prognosis, and may serve as a basis for using the CAVI index in secondary prevention programs. Furthermore, a study on Cardiovascular Prognostic Coupling is currently underway to determine the effect of baseline CAVI and changes in CAVI on cardiovascular events [35]. The study included 5109 consecutive patients with at least one cardiovascular risk factor. Follow-up is planned for ≥7 years, with annual CAVI determination. The primary outcome is the time to the onset of a serious cardiovascular event [35]. Data from this registry should provide information on the significance of baseline CAVI and changes in CAVI as indicators of cardiovascular prognosis in a representative patient population and serve as a basis for further research in this direction, including in the prediabetes patients.

There are several limitations to allow when considering the results of this study. First, there is a cross-sectional study design. Thus, the link between arterial stiffness and prediabetes cannot be considered a causal link. Second, in our study prediabetes was determined by as impaired fasting glucose or impaired glucose tolerance, or a combination of both, and oral glucose tolerance test and the assessment of HbA1c were not carried out. So, somewhat underestimate the real prevalence of prediabetes and type 2 diabetes, but in our opinion, this could not affect the association between prediabetes and the CAVI index that we identified, since in previously published studies it was the HbA1c index that was most associated with arterial stiffness [28], including in patients with prediabetes [20,21]; therefore, its definition could only strengthen the identified association, but not level it. Third, the population in this study was limited to Caucasian subjects living in one of the regions of Siberia; thus, the generalization of our results can be limited. Finally, we used body mass index to determine obesity, not the % body fat since this was the method provided by the ESSE-RF protocol. In addition, there are some caveats to consider when evaluating the clinical significance of CAVI. It is worth noting that CAVI measures the properties of the aorta, femoral artery, and tibial artery in general [12]. The aorta is an elastic vessel, and the femoral artery and tibial artery are muscle vessels under the control of nerves. Accordingly, the increased CAVI may represent not only vascular stiffness caused by pathological changes in the arterial wall but can also be explained by the increased vascular tone caused by smooth muscle contraction [36]. However, recent meta-analyzes and reviews have confirmed the less proven clinical and prognostic value of the CAVI index in a wide range of conditions that deserves further development into clinical practice.

## 5. Conclusions

In this study it was shown that in the population sample of the Russian Federation, the values of the CAVI index in patients with prediabetes and diabetes were higher than in the group of persons with normoglycemia (*p* < 0.001 and *p* < 0.009). Pathological values of CAVI (≥ 9.0) were more often detected in the groups of diabetes (16.8%) and pre-diabetes (15.9%) than in persons without CMD (9.0%). In the groups with normoglycemia and CMD various associations of indicators with CAVI were noted. Pathological CAVI in normoglycemia was determined by age, blood pressure, and dyslipidemia; in patients with CMD by blood pressure, visceral obesity, and decreased GFR. Since the identified disorders of arterial stiffness in prediabetes are similar to those in diabetes, their identification is important to prevent further cardiovascular complications. The possibility of using the CAVI index to monitor such preventive effects requires further research.

## Figures and Tables

**Figure 1 diagnostics-11-00474-f001:**
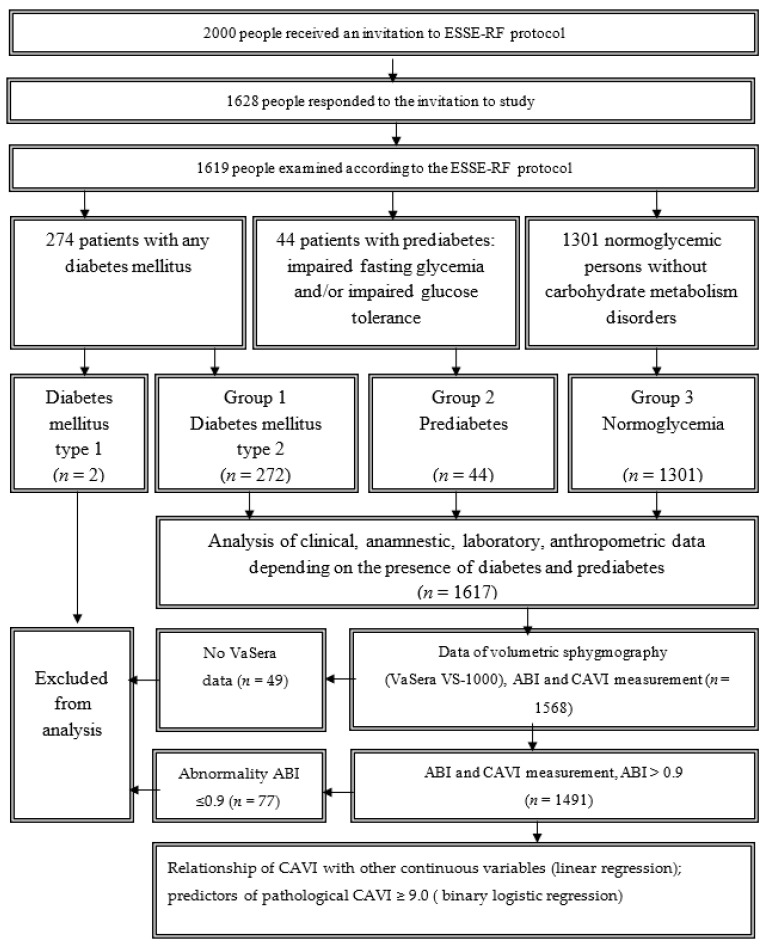
Flowchart of persons selection. CAVI: cardio-ankle vascular index.

**Figure 2 diagnostics-11-00474-f002:**
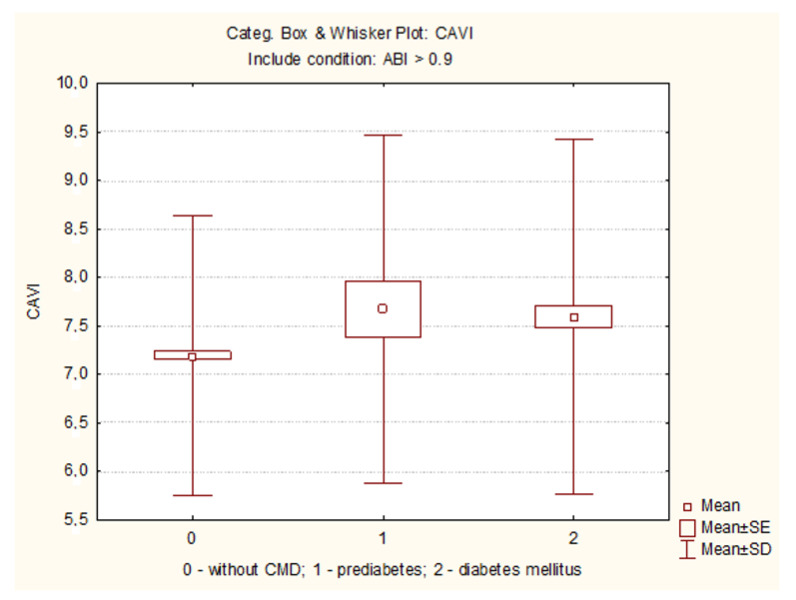
CAVI in groups with normoglycemia, prediabetes and diabetes. CAVI—cardio-ankle vascular index; *p* = 0.051 compared prediabetes to the normoglycemia group; *p* = 0.0002 compared diabetes to the normoglycemia group.

**Figure 3 diagnostics-11-00474-f003:**
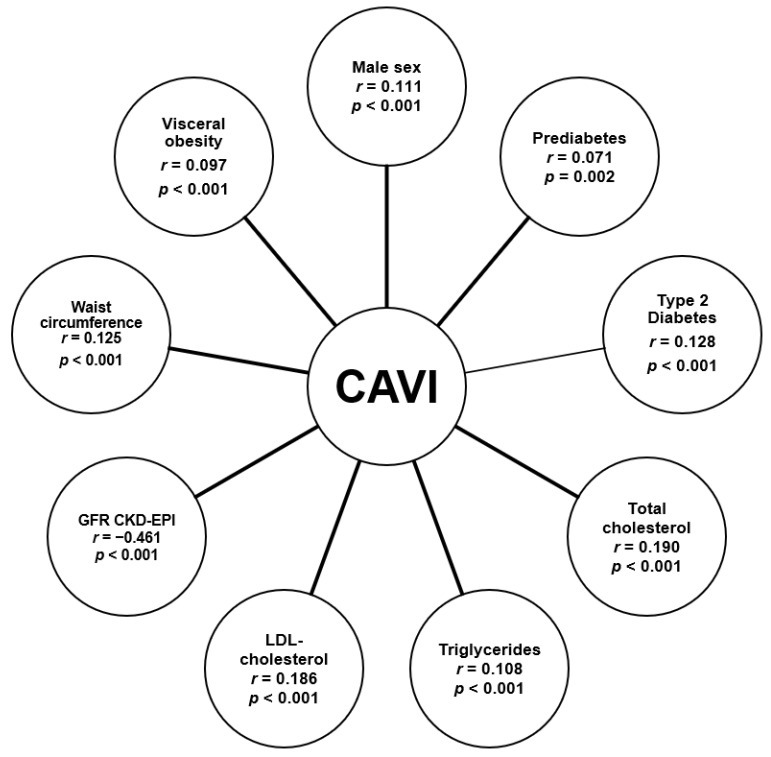
Correlation of CAVI with basic parameters in the total sample CAVI—cardio-ankle vascular index, GFR CKD-EPI—glomerular filtration rate calculated using the CKD-EPI formula, LDL cholesterol—low density lipoprotein cholesterol.

**Table 1 diagnostics-11-00474-t001:** Baseline characteristics of the study groups (*n* = 1617).

Variables	Group 1 Type 2 Diabetes *n* = 272	Group 2 Prediabetes *n* = 44	Group 3 Normoglycemia *n* = 1301	*p* Value ^a^
Age, years	55.0 (48.0;59;0)	52.5 (46.5;59.0) ^#^	46.0 (35.0;55.0) ^$^	<0.001
Male sex (*n*, %)	90 (33.1) *	23 (52.3)	584 (44.9) ^$^	0.002
Weight (kg)	88.1 (75.5;100.0)	84.85 (71.95;92.1) ^#^	75.8 (64.1;88.2) ^$^	<0.001
Height (cm)	165.6 (158.5;171.0)	167.6 (162.5;174.5) ^#^	168.0 (161.0;175.0)	0.002
Waist circumference (cm)	103.0 (90.1;112.0)	98.0 (91.0;107.5) ^#^	91.0 (81.0;101.0) ^$^	<0.001
Hip circumference (cm)	112.2 (103.0;120.0) *	106.0 (100.5;113.0) ^#^	102.0 (95.0;109.0) ^$^	<0.001
BMI (kg/m^2^)	32.9 (25.6;37.0)	30.1 (26.4;33.0) ^#^	26.7 (23.7;31.0) ^$^	<0.001
Obesity (BMI ≥ 30 kg/m^2^)	164 (60.3)	22 (50.0) ^#^	378 (29.1) ^$^	<0.001
Visceral obesity (waist circumference ≥ 80 cm in women and ≥94 cm in men)	228 (84.1)	35 (79.6)	809 (62.2) ^$^	<0.001
**Lifestyle, Physical Activity and Nutrition**				
Drinking alcohol more than once a week (*n*,%)	14 (5.15)	4 (9.1)	91 (7.0)	0. 346
Physical activity: sedentary work (*n*,%)	79 (29.0)	15 (34.1)	464 (35.7)	0.041
Time spent sitting in a day (hours, Me [LQ; UQ])	4.0 (3.0;7.0)	5.0 (3.0;9.0)	5.0 (3.0;8.0) ^$^	0.005
Total minutes of physical activity per day (minutes, Me [LQ; UQ])	240.0 (180.0;360.0)	240.0 (180.0;360.0)	300.0 (180.0;480.0) ^$^	0.007
Insufficient consumption of vegetables (*n*,%)	78 (28.7)	12 (27.3)	443 (34.1)	0.056
Insufficient consumption of fish and seafood (*n*,%)	48 (17.7)	343 (26.4)	9 (20.5) ^$^	0.006
Tobacco smoking (*n*,%)	64 (23.5)	10 (22.7) ^#^	419 (32.2) ^$^	0.006
Smoking experience (years)	36.5 (25.0;41.5)	33.0 (24.0;42.0) ^#^	24.0 (16.0;35.0) ^$^	<0.001
**Laboratory Data in Groups**				
Employed (*n*, %)	183 (67.3)	29 (65.9)	998 (76.7) ^$^	0.003
Arterial hypertension (*n*, %)	155 (57.0)	28 (63.6) ^#^	519 (39.9) ^$^	0.001
Ischemic heart disease (*n*, %)	47 (17.3)	5 (11.4)	90 (6.9) ^$^	0.001
History of stroke (*n*, %)	20 (1.5)	0 (0)	14 (5.15) ^$^	0.002
Kidney disease (*n*, %)	94 (34.6)	16 (36.4)	300 (23.1) ^$^	0.001
Glucose (mmol/L)	6.5 (5.0;7.0) *	6.3 (6.1;6.5) ^#^	4.8 (4.4;5.2) ^$^	<0.001
Total cholesterol (mmol/L)	5.4 (4.7;6.2)	5.7 (5.1;6.6) ^#^	5.1 (4.3;5.8) ^$^	<0.001
HDL cholesterol (mmol/L)	1.6 (1.3;1.9)	1.6 (1.4;1.9) ^#^	1.7 (1.4;2.0) ^$^	0.002
LDL cholesterol (mmol/L)	3.7 (3.0;4.2)	3.7 (3.2;4.6) ^#^	3.4 (2.7;4.1) ^$^	<0.001
Triglycerides (mmol/L)	1.6 (0.9;1.9)	1.6 (1.1;2.1) ^#^	1.2 (0.7;1.5) ^$^	<0.001
Creatinine (μmol/l)	69.4(63.5;76.9) *	73.45 (69.55;82.1) ^#^	69.5 (63.3;77.2)	0.003
GFR CKD-EPI (mL/min/ 1.73 m^2^)	88.8 (80.3;100.1)	96.6 (91.9;102.8) ^#^	103.0 (96.1;111.2) ^$^	<0.001
Uric acid (μmol/l)	0.3 (0.3;0.4)	0.4 (0.3;0.4) ^#^	0.3 (0.2;0.35) ^$^	<0.001
**Volume Sphygmography Data (VaSera VS-1000) *n* = 1586**				
	*n* = 268	*n* = 44	*n* = 1274	
Heart rate (beats/min)	76.8 (68.5;83.5)	76.5 (69.8;83.8) ^#^	73.0 (67.3;79.5) ^$^	<0.001
Systolic blood pressure-hand (mmHg)	137.75 (125.3;154.3)	134.75 (127.5;153.7) ^#^	128.5 (117.0;141.0) ^$^	<0.001
Diastolic blood pressure-hand (mmHg)	89.8 (82.0;96.0)	92.5 (81.5;101.0) ^#^	83.5 (75.5;92.5) ^$^	<0.001
CAVI	7.5 (6.7;8.6)	7.7 (6.7;8.4) ^#^	7.2 (6.3;7.9) ^$^	0.007
Pathological CAVI ≥ 9.0	45 (16.8)	7 (15.9)	114 (9.0) ^$^	<0.001
Conditionally pathological CAVI ≥ 8 (*n*, %)	108 (40.3)	17 (38.6)	297 (23.3) ^$^	<0.001

Continuous variables are presented as median with lower and upper quartiles (Me [LQ; UQ]). ^a^—results of three groups comparison according to Kruskal–Wallis test *—*p* < 0.017 when comparing groups 1 and 2. ^$^
*p* < 0.017 when comparing groups 2 and 3. *p* < 0.017 when comparing groups 1 and 3. BMI—body mass index, Visceral obesity—waist circumference ≥ 80 cm in women and ≥94 cm in men, HDL cholesterol—high density lipoprotein cholesterol, LDL cholesterol—low density lipoprotein cholesterol, GFR CKD-EPI—glomerular filtration rate calculated using the CKD-EPI formula, CAVI—cardio-ankle vascular index.

**Table 2 diagnostics-11-00474-t002:** Linear regression analysis (stepwise method) for the relationship of CAVI with other continuous variables (*n* = 1491).

	Unstandardized Coefficients		Standardized Coefficients	*t*	Sig. *p* Value	95.0% Confidence Interval for B	
	B	Std. Error	Beta			Lower Bound	Upper Bound
Total sample							
(Constant)	3.789	0.279		13.583	0.000	3.242	4.337
Age	0.058	0.003	0.432	16.761	0.000	0.052	0.065
BMI	−0.082	0.010	−0.336	−8.545	0.000	−0.100	−0.063
SBP	0.009	0.003	0.121	3.260	0.001	0.004	0.015
WC	0.011	0.004	0.106	2.575	0.010	0.003	0.020
DBP	0.009	0.004	0.080	2.204	0.028	0.001	0.018
Normoglycemia							
(Constant)	4.689	0.375		12.517	0.000	3.954	5.424
Age	0.056	0.004	0.440	15.778	0.000	0.049	0.063
BMI	−0.077	0.011	−0.312	−7.322	0.000	−0.098	−0.056
DBP	0.019	0.003	0.174	5.762	0.000	0.013	0.026
WC	0.012	0.005	0.114	2.536	0.011	0.003	0.021
HeartRate	−0.009	0.004	−0.059	−2.271	0.023	−0.016	−0.001
Type 2 diabetes and prediabetes							
(Constant)	3.536	0.770		4.592	0.000	2.020	5.052
Age	0.069	0.011	0.341	6.103	0.000	0.047	0.092
BMI	−0.085	0.015	−0.319	−5.779	0.000	−0.114	−0.056
SBP	0.023	0.005	0.260	4.508	0.000	0.013	0.033

The original model included: CAVI, Age, SBP, DBP, Heart Rate, WC, GFR_CKD-EPI, BMI, TG, Glucose, Cholesterol, LDL. BMI—body mass index, WC—waist circumference, HDL cholesterol—high density lipoprotein cholesterol, LDL—low density lipoprotein cholesterol, GFR CKD-EPI—glomerular filtration rate calculated using the CKD-EPI formula, CAVI—cardio-ankle vascular index.

**Table 3 diagnostics-11-00474-t003:** Results of binary logistic regression (forward LR method): association of factors with the presence of pathological CAVI (≥ 9.0) (*n* = 1491).

		B	S.E.	Wald	df	Sig. (*p* Value)	Exp(B)
	Total sample						
Step 1^a^	Age	0.102	0.011	80.959	1	0.000	1.108
	Constant	−7.430	0.632	138.339	1	0.000	0.001
Step 2^b^	Age	0.096	0.012	66.078	1	0.000	1.101
	DBP	0.035	0.007	22.664	1	0.000	1.035
	Constant	10.221	0.910	126.109	1	0.000	0.000
Step 3^c^	Age	0.101	0.012	70.339	1	0.000	1.106
	DBP	0.039	0.008	26.733	1	0.000	1.040
	BMI	−0.040	0.017	5.780	1	0.016	0.961
	Constant	−9.705	0.929	109.144	1	0.000	0.000
Step 4^d^	Age	0.099	0.012	68.207	1	0.000	1.104
	Stroke	0.905	0.447	4.104	1	0.043	2.471
	DBP	0.039	0.008	26.108	1	0.000	1.040
	BMI	−0.043	0.017	6.510	1	0.011	0.958
	Constant	−10.450	0.998	109.621	1	0.000	0.000
Normoglycemia							
Step 1^a^	Age	0.102	0.013	64.998	1	0.000	1.107
	Constant	−7.385	0.689	114.952	1	0.000	0.001
Step 2^b^	Age	0.095	0.013	52.678	1	0.000	1.100
	DBP	0.033	0.008	15.986	1	0.000	1.034
	Constant	−10.027	1.007	99.119	1	0.000	0.000
Step 3^c^	Age	0.101	0.013	56.932	1	0.000	1.106
	DBP	0.036	0.008	17.948	1	0.000	1.036
	Cholesterol	−0.225	0.098	5.280	1	0.022	0.799
	Constant	−9.344	1.041	80.604	1	0.000	0.000
	Type 2 diabetes and prediabetes						
Step 1^a^	SBP	0.030	0.008	14.699	1	0.000	1.031
	Constant	−6.122	1.183	26.782	1	0.000	0.002
Step 2^b^	SBP	0.023	0.008	7.944	1	0.005	1.023
	GFR_CKD-EPI	−0.077	0.027	8.203	1	0.004	0.926
	Constant	2.206	3.011	0.537	1	0.464	9.082
Step 3^c^	SBP	0.029	0.009	10.786	1	0.001	1.029
	GFR_CKD-EPI	−0.080	0.028	8.339	1	0.004	0.923
	Obesity	−0.834	0.388	4.623	1	0.032	0.435
	Constant	2.227	3.100	0.516	1	0.473	9.269
Step 4^d^	SBP	0.030	0.009	11.218	1	0.001	1.030
	Visceral Obesity	1.487	0.683	4.737	1	0.030	4.424
	GFR_CKD-EPI	−0.075	0.029	6.871	1	0.009	0.927
	Obesity	−1.264	0.421	9.038	1	0.003	0.282
	Constant	0.537	3.266	0.027	1	0.869	1.711

BMI—body mass index, SBP—Systolic blood pressure (hand), DBP—Diastolic blood pressure (hand), GFR CKD-EPI—glomerular filtration rate calculated using the CKD-EPI formula, CAVI—cardio-ankle vascular index.

## Data Availability

Data sharing not applicable.

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
