# Peer review of "Cardio-Ankle Vascular Index in the Persons with Pre-Diabetes and Diabetes Mellitus in the Population Sample of the Russian Federation"

_diagnostics, 2021, doi:10.3390/diagnostics11030474_

Round 1

Reviewer 1 Report

Sumin et al have demonstrated that arterial stiffness, as assessed through CAVI values, is elevated in pre-diabetes and diabetes patients compared to a normoglycemic control group. The concept of increased arterial stiffness in these patient groups has been described before (Loehr et al., 2016) however the previous study utilitsed more traditional methods (PWV) of assessing stiffness. Further, CAVI has been utilised to identify increased stiffness in pre-diabetic individuals (Namekata et al., 2016), albeit in a Japanese patient cohort. 
The merits of this study are therefore to confirm that the findings of the above work can be translated to a population of Russian subjects. 

Overall the work is robust, well described and well presented and is of interest. I have a few minor comments that could be addressed.

  1. It is not clear if there were a priori exclusion/inclusion criteria for this analysis. The authors have done a good job of transparently describing the data that was included/excluded but it is not clear if this was planned before the study
  2. Similarly, why were type 1 diabetes patients recruited in the first place if they were to be immediately excluded?
  3. A description of the typical racial makeup of the study participants will allow better contextualisation against the Japanese study mentioned above. 
  4. Why did the authors use BMI rather than % body fat for their determination of obesity? The limitations of BMI could be highlighted in the discussion.
  5. Is it true that all data in table 1 had a p value < 0.017 to be considered significant? Why this specific value?

Reviewer 2 Report

In this manuscript, the authors investigated CAVI and related cardiometabolic disorders in Russian subjects with normoglycemia, prediabetes, and diabetes. They observed that higher CAVI index values in prediabetic and diabetic individuals compared to those of normoglycemia. They also identified factors that are associated with higher CAVI index such as male sex, dyslipidemia, eGFR, and visceral obesity. Furthermore, CAVI index values of prediabetic and diabetic patients were similar, they concluded that their identifications are important to prevent further cardiovascular complications.

Comments:

1) As the authors mentioned, the reason why CVAI index values were similar in between normoglycemic subjects and prediabetic ones. Is it fair to interpret that hyperglycemia is not a risk for higher CAVI index? Please clarify this point.

2) Results are redundant and could be shorten. I suggest to write this part in several paragraphs.
